# Research on Wind Environment and Morphological Effects of High-Rise Buildings in Macau: An Example from the New Reclamation Area around Areia Preta

**DOI:** 10.3390/ijerph20054143

**Published:** 2023-02-25

**Authors:** Jialun He, Yile Chen, Liang Zheng, Jianyi Zheng

**Affiliations:** Faculty of Humanities and Arts, Macau University of Science and Technology, Taipa, Macau 999078, China

**Keywords:** high-rise and high-density urban wind environment, PHOENICS simulation, urban space form, high-rise residence, Macau Areia Preta

## Abstract

The Macau peninsula is close to the tropical ocean, with a high population density and a large number of high-rise buildings, which require a windy environment with good ventilation and heat dissipation. Based on residential samples and the degree of agglomeration, the high-rise residential area in Areia Preta was selected as the focus of this study. Meanwhile, summer typhoons pose serious safety risks to high-rise buildings. Therefore, it is necessary to study the connection between spatial form and the wind environment. First of all, this research is based on relevant concepts and the wind environment evaluation system of high-rise buildings and conducts research on high-rise residential areas in Areia Preta. PHOENICS software is used to simulate the prevailing monsoon in winter and summer, as well as a typhoon in an extreme wind environment, and summarize the wind environment’s characteristics. Secondly, by comparing the parameter calculation and simulation results, the possible relationship between the causes of each wind field is studied. Finally, conclusions are drawn about the urban form and wind environment of the site, and corresponding control strategies are proposed to reduce the shielding effect between buildings and typhoon damage. It can be used as a theoretical basis and reference point for urban construction and high-rise building planning and layout.

## 1. Introduction

### 1.1. Research Background

Macau is located on the west bank of the bustling Pearl River Estuary. The original land area was small, and land reclamation resolved the tension between contemporary urban development and physical space. The construction of residential complexes in the Areia Preta reclamation area has been accelerated, with an emphasis on increasing development intensity, and the economic benefits have improved [1,2]. However, this has also lad to many environmental issues that have been neglected during the development process. At the same time, building complexes generate more heat and exhaust gases during operation and use and are vulnerable to wind-related environmental problems. In recent years, Macau has developed rapidly, the numbers of local and foreign residents have gradually increased, and the demand for living space and quantity has also skyrocketed. With the rapid expansion of residential areas, typical high-density urban forms around the world have emerged. The environmental problems brought about by the development of high-density residential areas are becoming more and more obvious.

### 1.2. Literature Review

The outdoor wind environment of a building has a substantial effect on pedestrian comfort, natural ventilation in residential areas, and energy efficiency [3,4,5]. Nowadays, the intensity of urban development and the number of buildings has increased, significantly blocking urban ventilation and exacerbating urban air pollution and the heat island effect under low wind speed conditions [6,7]. As a result, urban wind problems are becoming more prominent. Therefore, it is very important to plan reasonably, improve the urban wind environment, and meet the safety requirements of the living environment. At present, studies have mainly focused on the ventilation efficiency of a single building and the wind environment of a specific type of building layout [8,9,10,11,12,13]. For example, scholar Bo Hong studied the relationship between different building layout forms and the urban heat island effect and found that a reasonable building layout can improve the urban wind and heat environment [8,9]. Ratti C. conducted a study on the ventilation efficiency of six combined buildings with a certain volume ratio. He pointed out that the ventilation efficiency of panel building groups is the best, and the ventilation efficiency of courtyard and row buildings is the worst [10]. Hu Yidong studied the outdoor ventilation environment quality of three architectural layout forms in Shanghai and found that the point type’s outdoor ventilation environment quality is the best, followed by the determinant type, and finally the enclosed type [11]. Wang Zhenwu studied the comfort of the wind environment in the four architectural layout forms of determinant, staggered, oblique, and peripheral and found that the wind environment in the peripheral architectural layout is the worst [12]. Huang Yemei used computer-based quantitative simulation methods to study the outdoor wind environment quality of 10 common building layout forms among the three typical forms of residential buildings. He believes that the quality of the outdoor wind environment is best in the building point layout [13]. In addition, different scholars have also conducted discussions on wind environment simulation for different types of residential buildings [14,15,16,17,18,19]. For example, these include hotels in the historic district of Qingdao [14], buildings in villages and towns in Xuzhou [15], traditional houses of the Tujia people [16], high-rise residential areas in hot summer and cold winter areas [17], coastal residential areas [18], and traditional residential buildings in southern Hunan [19].

In summary, currently the research on wind environment mainly focuses on numerical simulation and application technology [20,21,22,23], architectural layout and spatial form optimization [21,24,25], and wind environment optimization and residential building case studies [8,9,10,11,12,13,14,15,16,17,18,19]. Most of the research that has been done so far looks at the wind environment in daily situations. It is also important to think about how the wind affects buildings when the weather is bad.

### 1.3. Problem Statement and Objectives

The northwest Pacific Ocean is home to 36% of the world’s typhoons. According to public information from the Macau Meteorological and Geophysical Bureau, as of the second quarter of 2021, the No. 3 typhoon signal (meaning the sustained wind speed is 41–62 km/h, which is one of Macau’s typhoon warning signals) and the more advanced No. 8 suspension times reached 143, accounting for 46% of the total suspension impact times in the preceding 30 years. The frequency of typhoons is high and the wind is strong every year. Macau was struck by Typhoon Hato in 2017, resulting in eight deaths, 153 injuries, and 405 accidents, including falling billboards, windows, and building exterior walls. The structure and circulation field of the typhoon determine its tendency to move to the northwest; Macau is located to the northwest of where the typhoon originated. In addition, it is one of the most densely populated regions in the world. Due to the high population density and building density, as well as the height of its buildings, typhoons cause significant damage in Macau. Macau is just one of the typical cities with such an extreme climate.

On-site measurements, wind tunnel experiments, and software simulations are the three most important methods for studying wind environment. When comparing software simulation to other research methods, the advantages of convenience and cost savings become more apparent, and the technique has become a standard. It is utilized extensively in planning and design, structural design, and urban wind field prediction, among other fields. In this study, the CPD method is used to analyze two different wind environments, daily and extreme, for high-rise buildings in the ultra-high-density city of Macau. Under the specific climatic conditions and urban space of Macau, it is of great theoretical significance to explore the wind environment and urban architectural forms under conventional and extreme climates. In practice, by quantitatively studying the current situation of the existing wind environment, the relationship between climate and spatial form can be understood. At the same time, this study provides an empirical research reference for high-rise residential areas planned in Macau in the future, to enable the design of buildings and urban structures that adapt to specific climatic conditions. This paper also gives some information and references to help set up a wind environment research system for Macau. Practitioners of relevant design institutes can also make use of this study for effective references.

## 2. Study Area and Methods

CFD simulation software generates results that are quick, straightforward, accurate, economical, user-friendly, and intuitive. This paper simulates and analyzes the external wind environment of high-rise residential buildings in Areia Preta using PHOENICS software (2003 version). The study examines the advantages and disadvantages of the wind environment for urban ventilation and outdoor activities. Various planning control indicators are utilized simultaneously for quantitative calculation and analysis, and the relationship between urban form parameters and wind environment is determined. Pay particular attention to the following three points:Determine how urban form influences the wind environment and investigate the characteristics and causes.Examining the potential typhoon damage to high-rise buildings and its causes by comparing the calculated standard value of wind load with the simulation results.Determine the relationship between wind environment parameters and urban design parameters, and propose control strategies to improve the urban wind environment.

### 2.1. Study Area

The primary focus of this study is the new reclamation area of Areia Preta (Hac Sa Wan) in Macau. It is located in the northeast of the Macau peninsula. The research object is approximately 162,000 square meters in size and belongs to a typical high-rise residential area developed with high density and intensity. There are eight high-rise residential areas in the city, including La Marina, the Residencia Macau, Villa de Mer, La Baie du Noble, EDF. Polytec Garden, EDF. Jardim Kong Fok Cheong, EDF. Jardim Kong Fok On, and La Cite (Figure 1).

### 2.2. Simulation Condition Setting

In this study, the specific simulation experiment analysis process is shown in Figure 2.

The researchers simplified the building facade and removed unnecessary architectural decorations and components. Then set the calculation area and divide the grid.

#### 2.2.1. Computational Regions and Mesh Delineation

(1)Calculation area configuration: moderately adjust the grid density distribution according to the distance from the building complex itself and determine the sizes of the X, Y, and Z axes, which are 490, 1960, and 300, respectively.(2)Grid magnification factor: specify the grid spacing; the minimum is 5 m in the central area and 20 m at the edge, and the magnification factor near the center is 1.5 (Figure 3).

#### 2.2.2. Field Survey Data Correction

According to the research on the influence of different sizes of urban ventilation corridors on urban ventilation, urban corridors require a minimum corridor width to affect the outdoor wind environment. The size of this threshold depends on urban and rural terrain, but is usually between 6 m and 12 m. In this study, 6 m is used as the width threshold. According to the confirmation of land parcel ownership by the Macau Cartography and Cadastre Bureau, the area is divided into nine areas, and their horizontal and vertical layouts are summarized and analyzed.

#### 2.2.3. Computational Boundary Condition Settings

(1)In the calculation model, the K-turbulence model is employed, and ‘GASES’ gas No. 0 is selected as the gas type.(2)Choose a calculation condition model. The KEMODE turbulence model is frequently employed in simulations of outdoor wind environments.(3)In the urban gradient wind, the wind is affected by the roughness of the ground below a height of 300 m. In the process of software simulation, the empirical formula can be used as the expression of the wind speed profile to calculate the wind speed at different heights:
(1)$$U(Z)=U_{s}\;{\left(Z/Z_{s}\right)}^{\alpha }$$
where U_S_ denotes the reference height, Z_s_ denotes the average wind speed, and U(Z) denotes the average wind speed at height Z. Then, set the parameters according to the Chinese mainland green building design standard: “code for green design of civil buildings” (The Chinese Academy of Building Sciences is responsible for the interpretation of specific technical content. Green design applicable to new construction, renovation, and expansion of civil buildings is the most authoritative sustainable design criterion in China). It uses the basic scale, technical requirements, and technical measures as the principles of sustainable design. The incoming wind direction in this paper is 0.3 in summer and 0.12 in winter, based on the actual situation of the new reclamation area in Areia Preta, and the gradient wind height is 550 m.(4)Input the wind parameters of the building’s environment into the simulated wind environment, including wind speed, wind direction, and wind surface roughness.

## 3. Daily Wind Environment Characteristics and Building Complex Morphological Analysis

### 3.1. Wind Environment Simulation at Two Altitudes: Pedestrian Height and Standard Height

The evaluation of pedestrian wind comfort necessitates a comprehensive analysis of the pedestrian wind environment surrounding the building. In addition to studying the pedestrian height of 1.5 m on the ground, residential areas in Macau typically install garden landscape clubs and other venues in front of podiums to create an outdoor space for resident-frequented public activities. Consequently, two heights are chosen to represent wind field performance: pedestrian height (1.5 m) and standard height (10 m). Using the compiled data, the average parameters of the daily wind environment are calculated. In the setting of simulation conditions, there are two kinds of working conditions. One is the S direction in summer, with a wind speed of 3.36 m/s. The other is the N direction in winter, with a wind speed of 3.93 m/s. Calculate after importing the parameters, finish the convergence process of the calculated data, and obtain the simulated value after entering a stable state via the multi-segment calculation domain.

The study selected the pedestrian height of 1.5 m above the ground around the building as the reference height for wind speed measurement. Based on this, the impact of the near-ground wind environment in the block space on pedestrian activities is evaluated. Most countries, however, now use wind speed at a standard height of 10 m as the benchmark for wind-resistant building and structure design. Most of the lower podiums in the new reclamation area of Areia Preta are 10 m above the ground. Due to Macau’s special development model and policy regulations, the development of residential plots usually has podium parts with high coverage and fewer floors. Many residential podiums in the Areia Preta plot have leisure clubs, landscape gardens, community rest areas, and other places. As public places are used outdoors in the living environment, it is also necessary to study pedestrian comfort. Therefore, at the heights of 1.5 m and 10 m, the wind parameters in winter and summer under daily conditions are input to obtain the vector diagram of wind field velocity, surface pressure, airflow direction, and other results.

The study found that the overall wind speed of the summer monsoon environment is relatively comfortable for the human body, there are few places with high pressure, and there is no turbulent flow in the wind direction index (refer to Appendix A for detailed simulation experiment process). However, some typical wind environment issues persist in high-density urban agglomerations with large areas of wind shadow and local strong winds: (1) There are large areas of weak wind and calm wind above the streets inside the plot and on the leeward side of buildings. There are some environmental issues such as poor ventilation and heat dissipation in the streets and podium houses beneath the slab buildings during the summer. (2) The bottom of the first batch of windward buildings in the direction of the incoming wind and the roof of the podium under the point layout have better ventilation effects. The wind speed is greater than 2.5 m/s, and the wind comfort of pedestrians is more comfortable. (3) In summer, the internal street wind pressure is dominated by a low pressure difference, and the external windward side is dominated by a high pressure difference. At the same time, the overall wind speed of the winter monsoon environment is higher than the average value in summer, and the north side of the block is the junction of land and sea, with lower surface roughness and higher incoming flow velocity and average density. Furthermore, because of the sea and land breeze effect, when the incoming wind passes over the sea in winter, evaporative cooling brings ventilation to the buildings on the plot while increasing humidity and room temperature loss. There are many areas with high differential pressure, which increases the energy consumption of buildings in winter.

### 3.2. Influence Analysis of Morphological Elements of a Building Complex in Areia Preta

#### 3.2.1. Analysis on the Control of Street Height-to-Width Ratio on Ventilation Channels in Areia Preta

The platform is an essential component of a building in terms of communication and traffic distribution between the first floor and the exterior space. The podium form of the buildings in Areia Preta is primarily based on the plot’s base, and the podium is oriented such that it faces the street. The impact of the podium on the wind field is governed by the aspect ratio of the street canyon. Calculate the height-to-width ratio of the primary ventilation streets under each prevailing wind by categorizing the typical streets in the high-rise residential district of Areia Preta (Table 1). In this study, twelve street locations that are significantly changed by summer and winter are chosen as measuring stations. Figure 4 shows where the measuring stations are located: at the air intake and the air outlet, as well as on both sides of the street. Aa, Cc, Ee, Hh, Nn, and Kk are the measuring points of the wind intake for each street, while Bb, Dd, Ff, Jj, Ii, and Mm are the measuring sites of the wind outflow. The wind speed ratio is defined as the Y-axis value = UA/Ua (the ratio of one side to the other), and each measurement point on one side is the wind speed change ratio.

(1)In Table 1, the wind speed ratio (that is, the range of wind speed change) is further from 1.00, indicating a greater probability of wind speed change at both measuring stations. Numerous turbulent winds and vortices are present, and the local wind speed is greater than that of the surrounding area, or the wind field is chaotic. The closer the ratio is to 1, the smaller the possibility of wind speed variations on both sides of the place, which may be in a calm wind or laminar wind region. At the same time, the link between the height-to-width ratio of streets No. 1, 2, and 3 and the wind speed ratio between the wind intake and wind outflow is decreasing. According to this region’s wind environment features, the ratio of street height-to-width is inversely related to the wind speed ratio. In the future, it will be required to regulate the height-to-width ratio of streets in similar urban planning projects in order to avoid huge wind speed disparities or quiet wind regions.(2)The aspect ratio increase inhibits the bottom airflow. There is no correlation reference value for aspect ratio because streets No. 4, 5, and 6 do not share the same location variable. The aspect ratio of the adjacent street (No. 5) on the leeward side of the Residencia Macau is 4.88:1, which is an extraordinarily high ratio. The wind vortex generated on the top enhances the static wind state near the base, limiting the natural ventilation of the middle and lower floors of the Residencia Macau and La Cite and resulting in poor air quality. Over street No. 4 at this moment, the aspect ratio is 3.31:1, the wind speed near the ground is increased, and the vertically created wind vortex is stronger than the near-surface vortex. When the winter monsoon hits the 4th, 5th, and 6th streets, the air outlets are situated at the intersections of the streets, facing more complex multidirectional incoming flows. This causes a greater wind speed ratio, which represents the complexity of the wind field at the intersection.

#### 3.2.2. Effect of Street Orientation on Areia Preta Ventilation Corridors

The orientation of the urban ventilation channel in the neighborhood of Areia Preta is determined by the roadway network, and the buildings grouped along the street define the urban canyon space. Areia Preta’s street layout is based on the predominant north–south wind direction, and the angle of the current air passage is around 48.2 degrees. By comparing the three potential street orientations in Areia Preta, it is determined if the current urban streets of high-rise residential districts in Areia Preta match urban ventilation criteria (Figure 5).

(1)When the orientation is 0°, the simulation reveals that the north–south street is the prevailing wind ventilation channel under the north–south wind, and the internal airflow is relatively active, flowing down the canyon space of the street. On the street in the inner center, however, there is a phenomenon of two airflows converging, hedging, and encountering a downhill rushing wind when going upwards, which impairs pedestrian comfort. Overall, the airflow inside is quicker than the wind speed at 45°, there is a big area with a breeze above 2.5 m per second, and the ventilation state is good. However, the streets between La Marina—the Residencia Macau and the Residencia Macau—Villa de Mer have pronounced wind funnel effects, and the wind speed fluctuates substantially, which has a pronounced short-term amplifying effect on the passing airflow. In addition, there is a significant area of severe wind greater than 5 m/s at the building’s exterior corner. In comparison to the 90° orientation, the strong wind area at the bottom corner of the podium is more apparent in the 45° orientation. When buildings are oriented at 90°, the same phenomena occur, but the wind funnel effect and corner effect are more pronounced. When facing at a 90°, there is more turbulence in the inner block, and wind from numerous directions converges and intertwines elaborately in the inner street. Positive 90° and positive 0° are factors associated with more unfavorable wind field environments because they result in a more chaotic wind field environment, which has a relatively bad effect on ventilation, pedestrian comfort, and bottom safety.(2)In the present 48.2° and 45° orientation simulations, after the entering wind makes contact with the building complex, its performance is relatively balanced in all respects, and there is no wind speed greater than 5 m/s overall. When the wind blows from the north at an orientation of 48.2°, the wind speed on the majority of streets is between 0 to 2.4 m/s, which is not conducive to the normal ventilation and heat dissipation within the building complex. In addition, it increases the area of the building’s wind shadow area, so it can be concluded that the present orientation of 40.3° is one of the causes of the building group effect. This simulation demonstrates that when the alignment of the roadway is completely parallel to the direction of the incoming wind, a uniform laminar airflow with an increasing wind speed with height is produced. However, this will enhance the likelihood of sudden gusts of wind, and the street corner will have a greater concentration of strong winds. At this moment, improvements can be made by enhancing the form and aspect ratio of the street plan. When the windward angle of the building is close to 45° to the prevailing wind direction, the problem of internal ventilation and structures blocking the downwind direction may be well solved, and the effective span of the windward side of the building group is considerably different. The prevailing wind throughout the year decreases to create a uniformly distributed outside wind field. Therefore, the high-rise residential neighborhood in Areia Preta has a good ventilation penetration effect in the face of the yearly predominant north wind, and the downwind direction and local streets can be better ventilated. As a result of the pattern spacing of high-rise structures, there are still small wind dead zones in streets perpendicular to the direction of incoming wind. In conjunction with the control indicators of the street height-to-width ratio and line-pasting rate, a thorough investigation and evaluation are undertaken, and the urban form control strategy for future urban design in similar situations may be established.

#### 3.2.3. Building Complex Frontal Region

The building frontal area ratio reflects the dimensions of urban spatial form’s influence on air circulation. Using the north–south prevailing wind direction of the high-rise buildings in Areia Preta as a reference, Villa de Mer, the Residencia Macau, and La Marina comprise the first row of windward structures positioned in the north prevailing wind. Figure 6 shows that the first row of buildings facing the south wind are EDF. Polytec Garden, EDF. Jardim Kong Fok Cheong, EDF. Jardim Kong Fok On, La Baie du Noble, and Villa de Mer.

The area of the south windward side is approximately 48,688.98 m^2^, and the area of the north windward side is approximately 61,842.56 m^2^. The calculation shows that the ratio of windward area is 0.518 on the south side and 0.658 on the north side. According to the simulation analysis of the wind environment under the aforementioned daily working conditions, a greater windward area ratio results in a stronger screen building impact. For instance, the narrow pipe effect created by the close proximity of buildings in Villa de Mer increases wind speed by two to four times. In addition to the positive effect of the urban ventilation channel in Areia Preta, the tall facade wind barrier has created a significant urban wind shadow region within the building complex. In the simulation of summer working conditions, where the internal enclosure, building density, and average building height of the area are assumed to be quantitative, a smaller windward area ratio results in greater wind penetration within Areia Preta. For non-parallel-flow roadways, the wind speed within a particular range is also increased. It can be shown that the ratio of the windward area of the building complex in Areia Preta is one of the most influential elements of the building complex.

#### 3.2.4. Arrangement of Building Heights

The degree of building arrangement is expressed by the index of strewnness, which indicates the floating level of height difference in the building group and the varying levels of buildings on the plot in the vertical space. In the study, the average level of displacement in each residential neighborhood and all high-rise buildings in Areia Preta was determined. The average height of buildings in the high-rise residential neighborhood of Areia Preta is 118.75 m, with La Marina, the Residencia Macau, Villa de Mer, and La Cite having a staggered degree of +30.25 m. La Baie du Noble is +19.25 m, EDF. Polytec Garden is +1.25 m, and EDF. Jardim Kong Fok Cheong and EDF. Jardim Kong Fok On are −70.75 m. The summertime due south wind field environment is chosen as the research reference (Figure 7). When the staggered degree is low, the building height level is lower and at the same height, and the skimming airflow can continue to flow in the original wind direction. When the impact of the degree of randomness is greater and the height of the building is considerable, as the height of the local building increases, the local wind speed at the corner of the building increases, and the airflow flows along the sides of the building. The more staggered the buildings, the higher the wind speed near the ground, the slower the flying airflow, and the better the shielding function of the building complex.

#### 3.2.5. The Influence of Group Combination on the Formation of Monomers

Under the group combination of buildings, the wind environment is compared, and the placement of the measuring sites is also specified in the group simulation in order to further assess how the group combination has affected the Areia Preta high−rise building. At sites A3, B3, B4, and C4, the study found that the differences in wind speed between the group simulation and the individual simulation are relatively considerable (Figure 8). The building corner is something that all of these sites have in common. This corner is made when the buildings in a group work together, which is very different from the simulation of a single building (refer to Appendix B for the detailed simulation experiment process). The arrangement of building groups has a significant impact on the wind field distribution. Comparing the individual wind fields reveals that the wind speed increases significantly at the corners of the U−shaped aircraft hangar on both sides. Affected by neighboring buildings and street space, it is approximately 0.5–1 times larger than when it exists on its own (red circle in Figure 9). When carrying out urban design, consideration must be given to the increase in wind speed at the corners of the windward side of high−rise buildings, especially if there are other high−rise buildings on both sides, as the wind speed increase multiplier will be greater than 0.5. The shading between buildings decreases ventilation in areas that would otherwise have it. The L−shaped slab building is located within the building complex, and the majority of it is in the wind shadow area of the front row of buildings when it is obstructed by the front row of buildings. The wind velocity is less than that of the single−body simulation, which accentuates the phenomenon of stillness and absence of wind in the enclosed space (Figure 9, white circle). As the building’s corner unit, A3 has a good ventilation effect, and the wind speed is approximately one-sixth that of a single unit due to the front row’s shading (Figure 9, the black connection). The interaction between single buildings also affects point-type single buildings. The distance between point−type buildings is small, the flow effect of the surrounding wind field is less than that of the single-body simulation, and the wind speed at the corner is approximately one−fifth of its original value (Figure 9, the red connection).

By analyzing various urban form indicators, the causes of wind environment problems in Areia Preta’s buildings and how to induce their effects are determined. The following conclusions were reached:Due to the large aspect ratio of the building itself, the degree of enclosure, the large aspect ratio of the building groups, and the windward area ratio, as well as other morphological factors, the surface wind speed near the inner streets and podium buildings in winter and summer is greater than that of the outer streets. These are commonly low. A large area of wind shadow and vortex areas appeared in the sky, which weakened the ventilation efficiency and led to a poor ventilation environment within the high-rise buildings of Areia Preta.The region of highest positive pressure appears on the leeward side of Winter Sea Residence and Grand Hyatt Bay. The narrow spacing between the former building layout and the 15° orientation of the tower exacerbates the narrow tube effect, thereby doubling or tripling the wind velocity. Because it is U−shaped, the latter is good at spreading out the incoming wind in the winter, which increases the airflow in the back and lowers the pressure difference caused by the negative pressure in the back.At the opening of the exterior block and the building’s corner, the wind velocity is greatest. Due to the angle of deflection between the street orientation and wind direction, there is a corner effect at the intersection and street opening, which increases the local wind speed.Because of building shading, the L−shaped plate building’s overhead floor does not help much with ventilation and heat loss, and the vortex wind has a lot of trouble with the podium roof.

## 4. Analysis of Potential Damage Caused by Typhoon Weather

Due to Macau’s unique geographical location, typhoon-caused severe weather is a nuisance and a common cause of property damage. During the period of 7–17 September 2018, for instance, it experienced a super typhoon that had not occurred in half a century, and the strongest winds reached level 14. Most typhoons originate from the subtropical high in the western Pacific Ocean. According to the geostrophic force of the northern hemisphere and the path of Typhoon Mangkhut, the average wind direction in Macau is predominantly easterly or southeasterly when a typhoon lands along the coast. During the “Mangosteen” typhoon, for instance, the stable wind in the Areia Preta was 32.7 m/s and the fluctuating wind was 52.3 m/s.

### 4.1. Analysis of Typhoon Simulation Environment Results

The wind speed distribution map at heights of 1.5 m, 20 m, 50 m, and 100 m, the three-dimensional cloud map, and the pressure wind speed map on the building surface are generated by inputting the instantaneous fluctuating wind value during the typhoon. Villa de Mer, La Baie du Noble, and EDF. Polytec Garden were identified as the three wind−facing building complexes in Areia Preta. At the corner of the high-rise building, a maximum instantaneous wind speed of 75 m/s is formed. The La Baie du Noble and EDF. Polytec Garden buildings are obstructed by point-type towers in the front row, and the maximum wind speed of the structures in the back row falls below 60 m/s.

The windward side of the building complex is taller than the leeward side, and the leeward side of EDF. Jardim Kong Fok Cheong, EDF. Jardim Kong Fok On, and school buildings have more wind-sheltered areas. The southeast–northwest street wind speed on Rua Central da Areia Preta exceeds 20 m/s, with the exception of a few spots at the back of slab buildings, making pedestrian activities extremely hazardous. At the corner under the building podium on the windward side, winds in excess of 75 m/s are detected. There are 10 m, 50 m, and 100 m canyon winds between Villa de Mer and La Baie du Noble, and La Baie du Noble and EDF. Polytec Garden. Furthermore, as the height increases, the speed expansion coefficient increases, and the maximum speed at a height of 50 m exceeds 80 m/s. The internal streets are obstructed by tall buildings, and the wind speed in a large portion of the city is less than 20 m/s, which can severely impair the travel comfort of the inhabitants. On the windward side of the building, the pressure surface gradually increases from about 2000 Pa to about 3000 Pa. On the leeward side of Villa de Mer, the value range is approximately 400 Pa to −900 Pa, and the pressure difference is relatively large. On the leeward side of La Baie du Noble, the range of values is about −900 Pa to −350 Pa. On the leeward side of EDF. Polytec Garden, the range of values is about −100 Pa to −600 Pa (refer to Appendix C for detailed simulation experiment process).

### 4.2. Analysis of Vulnerable Parts of Building Monolith and Standard Floor Plane

On the windward side of the buildings in this study, the Villa de Mer, La Baie du Noble, and EDF. Polytec Garden during Typhoon Mangkhut were selected, and the variable wind values were used to calculate the standard values of downwind loads at each height. Among the three residential areas, the building with the highest value is chosen, and the Tecpolt software (RS version) is used for CFD simulation post−processing. Set the points between 1.5 and 100 m in height, determine the specific pressure value at each point, and calculate the average wind pressure. Comparing the above calculation results with CFD typhoon simulations reveals that the majority of the standard values of wind loads that occur once every 50 years are less than the pressure value of the windward side of the first row of buildings during Typhoon Mangkhut. The residential area of Villa de Mer is dominated by five point towers, the standard floor plan consists of two elevators and seven dwellings, and the first floor is a raised pillar floor. Under the width of the windward side during Typhoon Mangkhut, the instances of 50 and 100 m exceeded the standard value of wind load by approximately 1.14 and 1.22 times, respectively, if the larger average value is used. Under the width of B’s windward side, the multiples of 50 and 100 m exceed the standard wind load value by approximately 1.34 and 1.56, respectively. Consequently, it is determined that the arc-shaped balcony in the longitudinal apartment type and the corner window in the living room in the horizontal apartment type are the plane’s weakest points. At a height of 100 m and above, the wind pressure on the windward doors and windows exceeds 1.2 times the maximum width of the front face, and the requirements for the wind pressure resistance of doors and windows are more stringent than those for the low-rise areas of the same building (refer to Appendix C for detailed simulation experiment process).

The residential area of La Baie du Noble is dominated by five slab buildings; the standard floor has a plan layout with two elevators and four households, while the first and 14th floors are elevated pillar floors. Under the width of the windward side during the Mangkhut typhoon, the instances of 50 and 100 m exceeded the standard value of wind load by about 1.77 and 1.95 times, respectively, the larger of the two average values. The side windows of the unit type on the windward side and the curved balcony in the southwest (Figure 10b, red frame) are deemed to be vulnerable components in the standard floor plan of the building type. According to the post-disaster images of wind disaster records in Figure 10d, the distribution of window damage is concentrated in the bedroom windows between the curved balconies, while the relatively small side windows on both sides have relatively less damage, which is largely consistent with the calculated results.

The residential area of EDF. Polytec Garden is dominated by six point-style towers; the standard floor consists of two elevators and seven dwellings; and the first floor is an elevated column floor. Under the width of the windward side of a during Typhoon Mangkhut, the times of 10, 50, and 100 m exceed the standard value of wind load by about 1.25, 1.09, and 1.04 times, respectively. Under the width of B’s windward side, 10 m, 50 m, and 100 m, respectively, exceed the standard value of wind load by about 1.68, 1.40, and 1.38 times. Therefore, it is determined that the concave corner of the cross plane forms a relatively high numerical wind pressure and a relatively low wind speed forms a vortex wind area, resulting in a funnel effect. Compared to the 90° windward angle of La Baie du Noble, the 45° windward angle of the EDF. Polytec Garden Building effectively eliminates wind speed; however, the vertical wind poses a risk of window damage from ground-borne projectiles. Most of the time, the coastline starts out as open water, has smooth ground, and directly blocks houses that are closer to the typhoon’s path. This increases surface pressure and the risk of damage to door and window parts. When a powerful typhoon approaches, residents must be cautious in the aforementioned indoor areas. Due to the vast pressure difference between indoor and outdoor areas, door and window components are likely to break or to fail to close due to insufficient design bearing capacity.

### 4.3. Analysis of Building Groups and Vulnerable Parts of Facades

This study examines the simulated values of high-rise buildings in extreme typhoon conditions using the PHOENICS software. Consistent with the previous section, it mainly discusses three residential areas on the windward side, namely, Villa de Mer, La Baie du Noble, and EDF. Polytec Garden. Figure 11a shows how to look at the X-axis slices of the buildings in each residential area and analyze the wind speed and wind pressure vector conditions inside the buildings.

Observing the simulation results of each building reveals that when the incoming wind blows on the building, a wind vortex forms at the base of the windward side of the buildings in the three residential areas, and the wind speed decreases within the wind vortex. The building’s rear creates a certain range of wind shadows. The eaves at the top of the building become the location with the highest wind speed; the maximum at the top eaves of La Baie du Noble is 65 m/s, while the maximum at the top eaves of EDF. Polytec Garden and La Baie du Noble is 60 m/s. The facades of the high-rise residential units in the three residential areas all have positive wind pressure peak areas that are close to negative pressure peak areas on the roof, which can easily lead to excessive suction of the door and window components. Consequently, parapet walls can be added to the edge of the roof during design or subsequent renovation in order to lift the vortex from the roof surface. Or install a protrusion or louvered spoiler at the corner to eliminate the negative pressure peak and separate the vortex. The entrances to the residential buildings in this area are designed to lead directly to the podium roof on the first floor. Therefore, the rain shield above the entrance door must increase its load-bearing capacity and structural integrity to accommodate the increased precipitation brought on by high winds, moving air, and falling objects. The podium can effectively mitigate the narrow pipe effect and the impact of the vortex downdraft on pedestrians in the streets of high-rise buildings. The podium structure may be used to block the downdraft, and protective ceilings may be installed at the building’s primary entrances and exits. As shown in Figure 11b, the simulation results of Villa de Mer indicate that the areas with the highest surface wind velocity are the eaves, roof corners, and elevator machine room, where local damage is likely to occur; the surface wind velocity of the main facade is low and inversely proportional to its height. Planarly, EDF. Polytec Garden resembles Villa de Mer, and the peak wind speed distribution area resembles Villa de Mer. The simulation results from La Baie du Noble demonstrate that the wind speed is greatest at the L-shaped slab building’s corner and is inversely proportional to its height. The maximum wind speed of the third building in La Baie du Noble exceeds 70 m/s at the top and 50 m/s at the base. Incoming wind speed decreases gradually as it passes through the building surface, while corner effect wind speed increases locally. Due to the fact that the plan form of La Baie du Noble is T4, there is a large air shaft between the front and back rows of units (Figure 11b, the red box). The front row windward unit has a shallower depth than other residential areas, and its high-rise units are more likely to cause a significant difference between indoor and outdoor wind pressure, leading to excessive suction and damage to doors, windows, curtain walls, and other maintenance components.

## 5. Conclusions

This paper’s research is based on computer simulation methods. The wind field of the dominant monsoon in winter and summer on high-rise buildings in Areia Preta was simulated using PHOENICS software, and the wind environment conditions under daily working conditions were analyzed. It is also concluded that the aspect ratio, enclosure degree, building orientation, street height–width ratio, street orientation, windward area ratio, and staggered degree of individual buildings all have a significant impact on the wind environment of the buildings in Areia Preta, as well as how to mitigate the group effect that resulted in high-density high-rise buildings. During the process of analyzing wind field problems and influencing factors, some improvement methods and strategies were also summarized: (1) Strictly control building scale; further, it has been discovered that the height-to-width ratio of buildings in this area has a greater impact than the aspect ratio. The shielding effect of buildings on the city can be reduced by appropriately increasing the ventilation rate of high-rise buildings, such as overhead ratio, facade permeability, and air shaft location, among other things. (2) Avoid using an L-shaped layout when the plate plane is too continuous. For the dotted layout, a staggered layout is recommended. More attention should be paid in the overall layout to the reservation of the air passage for the incoming wind direction, in order to meet the ventilation needs of buildings and urban areas in various layouts. (3) Slab buildings should be oriented 45–75° east by south, while point buildings should be oriented 0 or 30° east by south. (4) Strictly control the street height-to-width ratio, which should be kept between 1–3, but not so large that it interferes with comfort. (5) A 45° angle between the street and the prevailing wind can help to maintain a good urban ventilation channel effect. (6) Components such as doors, windows, and curtain walls on the typhoon’s windward side must have greater load-carrying capacity to reduce the risk of typhoon penetration on the opposite side and buildings. (7) To reduce the damage caused by high-intensity typhoons to the building facade enclosure structure, low walls, wind deflectors, and other components should be installed on the roofs of high-rise buildings.

In the event of a typhoon, the windward side buildings and the city’s high wind speed and high pressure distribution areas are analyzed by comparing the software simulation results with the standard value of wind load. The wind load standard values set for almost all parts of the housing on the windward side are less than the wind pressure caused by the Mangkhut typhoon’s instantaneous strong wind. In contrast to the daily simulation, the opening of the building overhead layer allows some of the concentrated wind pressure to escape. Maximum wind speed occurs at the building’s eaves and corners; maximum wind speed near the ground occurs between the north and south of the two buildings, as well as at the street entrance.

In general, the planning and design of high-rise residential buildings must not only maximize high economic benefits. When ensuring the functions of buildings and cities, we should take into account the urban wind environment in order to ensure normal ventilation and heat dissipation in the living environment, as well as the comfort and safety of outdoor activities near pedestrian height on the ground. When facing the planning of high-rise buildings in the new reclamation area of Macau, it is also necessary to consider the frequent occurrence of typhoons. This study can be used as a starting point for a preliminary analysis of the wind environment planning for high-rise residential buildings in cities with a lot of people. In the future, on-site measurements and wind tunnel experiments can be used together with more in-depth research on specific topics.

## Figures and Tables

**Figure 1 ijerph-20-04143-f001:**
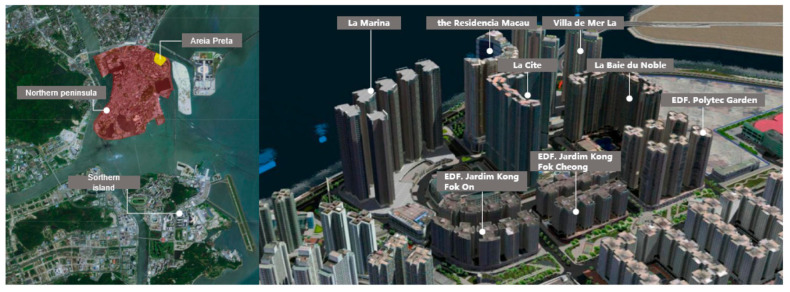
Study Area.

**Figure 2 ijerph-20-04143-f002:**
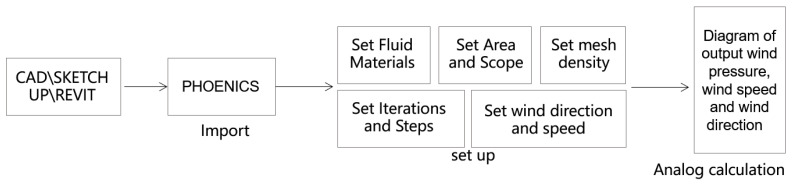
Simulation analysis process.

**Figure 3 ijerph-20-04143-f003:**
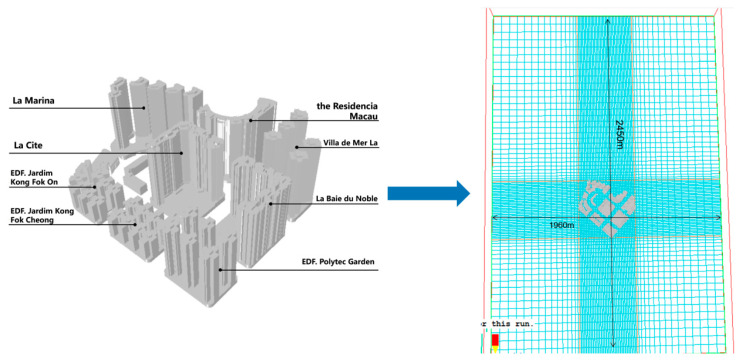
The optimized model and mesh enlargement.

**Figure 4 ijerph-20-04143-f004:**
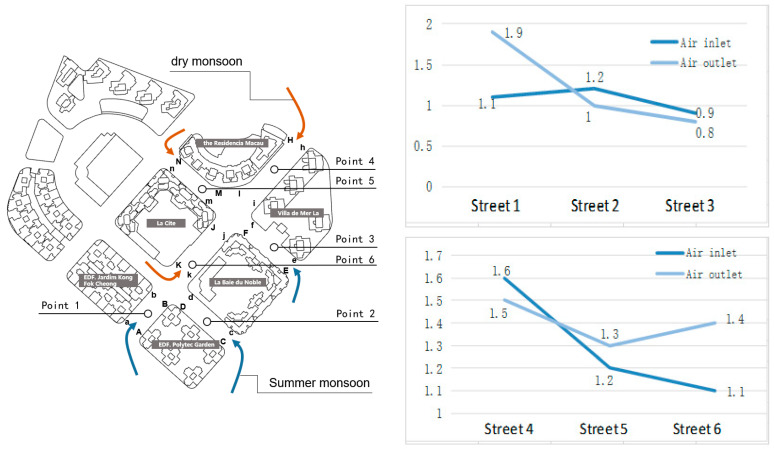
Location of incident flow wind distribution of measuring points and streets (blue is summer monsoon, red is winter monsoon); the change ratio of wind speed in each street during the summer monsoon (street 1 to 3) and winter monsoon (street 4 to 6).

**Figure 5 ijerph-20-04143-f005:**
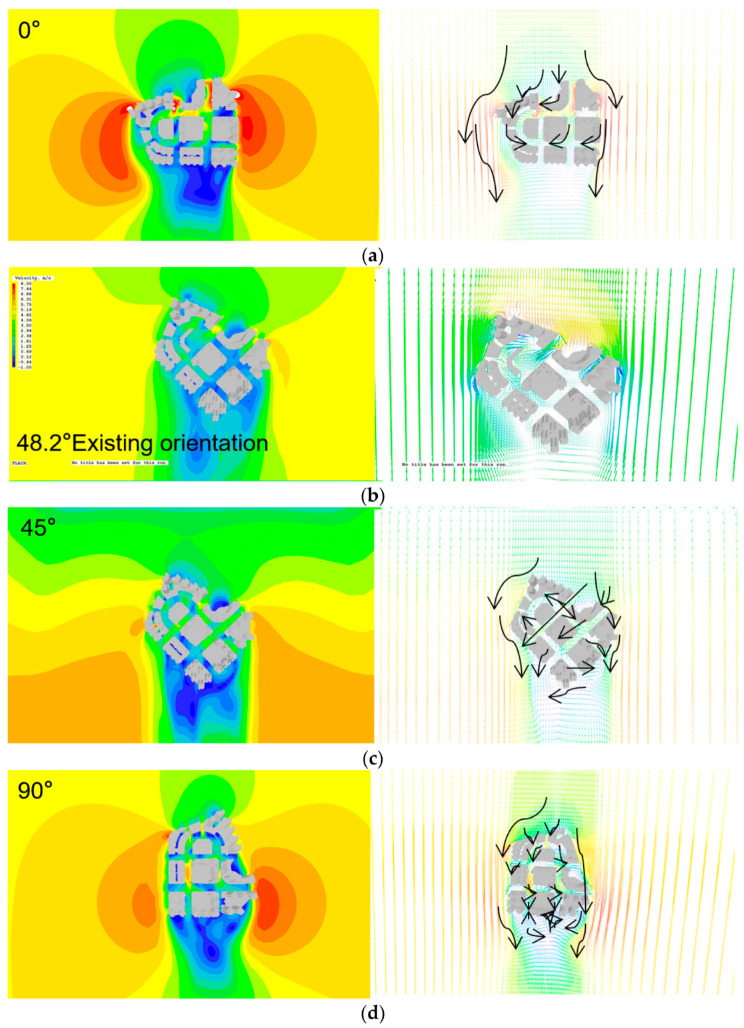
Results of the analysis of wind direction and street orientation. (**a**) Schematic diagram of wind direction and street orientation at 0°; (**b**) Schematic diagram of wind direction and street orientation at 48.2°; (**c**) Schematic diagram of wind direction and street orientation at 45°; (**d**) Schematic diagram of wind direction and street orientation at 90°.

**Figure 6 ijerph-20-04143-f006:**
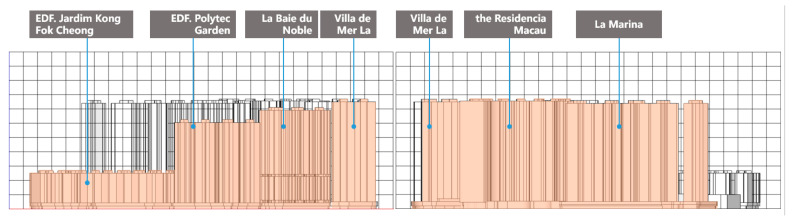
Schematic diagram of the area of the windward side ((**left**): south wind direction, (**right**): north wind direction).

**Figure 7 ijerph-20-04143-f007:**
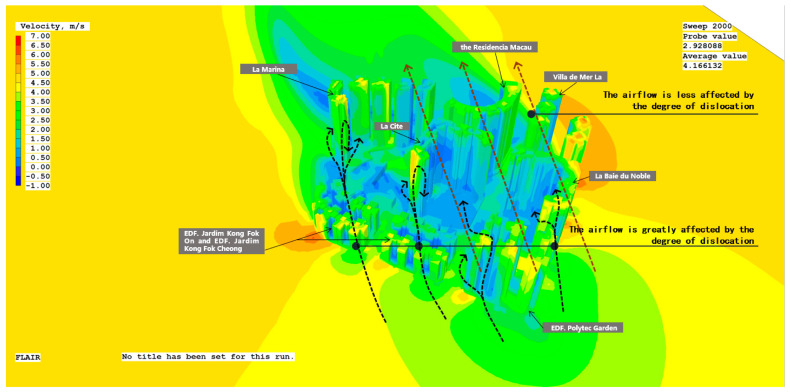
Surface wind speed cloud map of summer monsoon simulation. (Image source: drawn by the author).

**Figure 8 ijerph-20-04143-f008:**
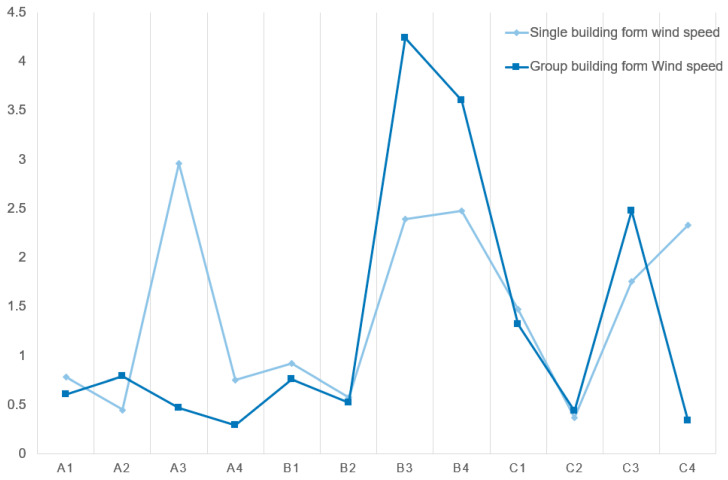
Comparison of wind speed at the same point between swarm simulation and individual simulation.

**Figure 9 ijerph-20-04143-f009:**
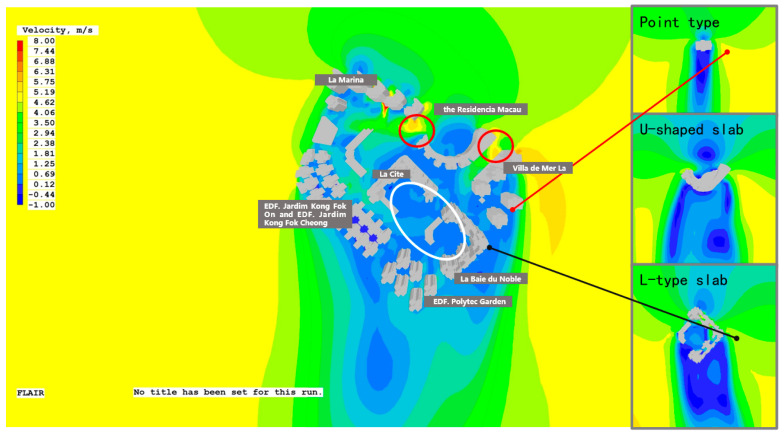
Wind speed performance of building complexes in Areia Preta.

**Figure 10 ijerph-20-04143-f010:**
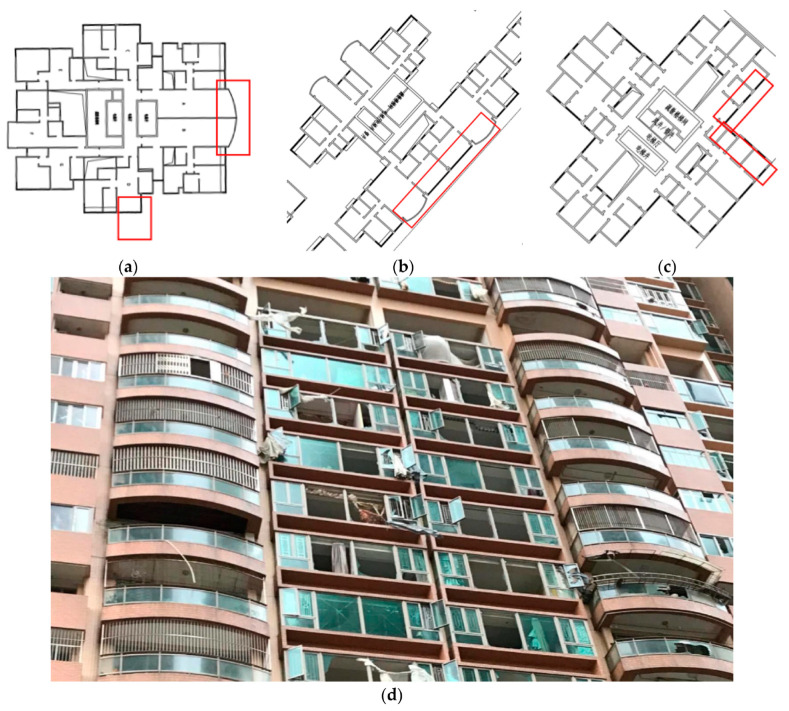
Locations vulnerable to typhoon damage in the standard layer plane. (**a**) Villa de Mer standard floor; (**b**) La Baie du Noble standard floor; (**c**) EDF. Polytec Garden standard floor; and (**d**) Damage to the facade windows of La Baie du Noble after Typhoon Hato. In the floor plan (**a**–**c**), the Chinese text indicates the location of the elevator room of the residence.

**Figure 11 ijerph-20-04143-f011:**
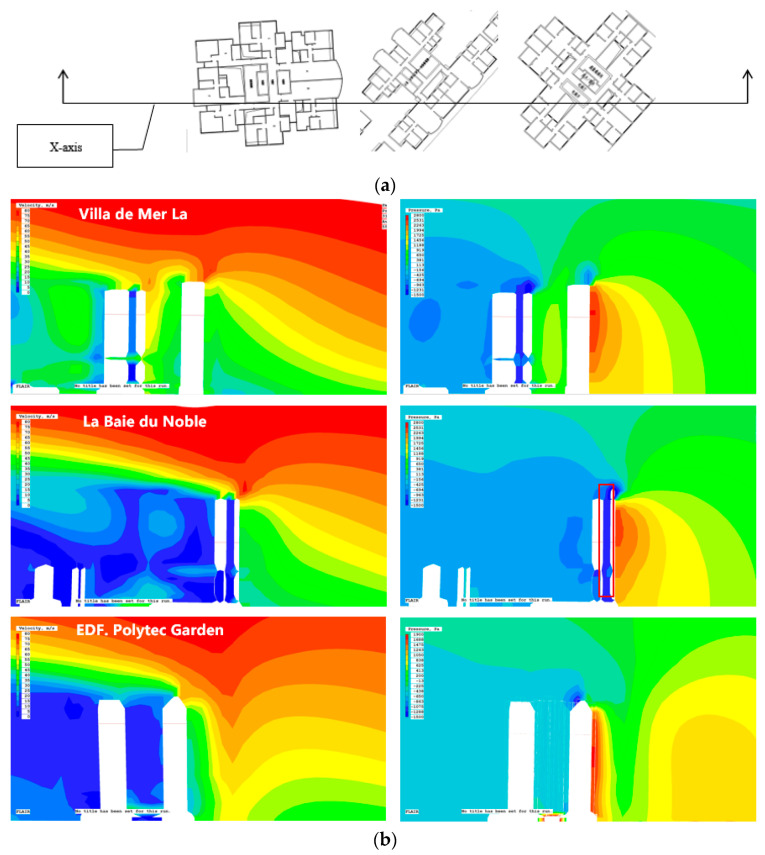
(**a**) X-axis slice position top view; and (**b**) Cloud chart of X-axis section of wind speed and wind pressure nephogram. In the floor plan (**a**), the Chinese text indicates the location of the elevator room of the residence.

**Table 1 ijerph-20-04143-t001:** The average value of the height-to-width ratio of each street that is greatly affected by monsoons.

Effect of the Wind Direction	Set Serial Number	Representative Street Segments	Street Length	Street Aspect Ratios
prevailing wind in summer	1	EDF. Jardim Kong Fok Cheong—EDF. Polytec Garden Section, Oriental Pearl Street	78 m	2.32:1
	2	EDF. Polytec Garden—La Baie Du Noble Section, Rua Central da Areia Preta	111 m	2.50:1
	3	Villa de Mer—La Baie Du Noble section, Rua Central da Areia Preta	111 m	2.52:1
prevailing wind in winter	4	The Residencia Macau—Villa de Mer section	138 m	3.29:1
	5	The Residencia Macau—La Cite Section, Rua Central da Areia Preta	111 m	4.80:1
	6	La Cite—La Baie Du Noble Section, Oriental Pearl Street	114 m	1.47:1

## Data Availability

The datasets used and analyzed during the current study are available from the corresponding author on reasonable request.

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
