# Peer review of "Research on Wind Environment and Morphological Effects of High-Rise Buildings in Macau: An Example from the New Reclamation Area around Areia Preta"

_ijerph, 2023, doi:10.3390/ijerph20054143_

Round 1
Reviewer 1 Report
Dear authors,
the work is well presented. However, I see some issues:
1. abstract I suggest to follow 1-2 sentences for general context, 1-2 for methodologies, 1-2 for results, 1-2 main implications
2. the section 1 is weak. I suggest to propose main concepts, the gap of your research. If you want, you can propose a specific section of literature analysis
3. methodology. what is the comparison with existing literatures? please justify the choices
4. the choice of input data, please provided details
5. I see a very interesting interpretation of results, please provided a comparison with existing literature when it is possible
6. I suggest to reduce redundant concepts, in some cases you can transfer them also in supplementary file
7. I suggest to analyze some aspects on social side (recent literature): i. stakeholders engagement; ii. sustainable education. I think that can be improved the system.
Author Response
Response: Thank you for your suggestion.
(1)Our Abstract section has been rewritten. Sentences are condensed than before. Currently 210 words. Structurally references your proposal as well. For details you can see the modified documentation.
(2) (5) (7) In the first verse, we retuned. The background of the research is built, pointing out the practical needs and research purposes of our research. Specific literature review analysis sections have also been adjusted. Most of the existing papers analyze the daily wind environment. This paper analyzes the parameters of the daily and extreme wind environments, analyzes the causes of the respective wind fields, explores the potential relationship, and reveals the correlation between urban form and wind environment. It mainly takes the related concepts and evaluation system of wind environment of high-rise buildings as the theoretical basis.
(3) The research methods are mainly computer numerical simulation, parameter analysis method and survey research method. However, our wind environment analysis is the focus of this article, so it is mainly based on computer numerical simulation experiments. The survey research is the data collection in the early stage and the verification after the experiment. The details correspond to the second part of the article. We have adjusted some lengthy sentences.
(4)For the data mentioned in 2.2.4(4) of the article, the ground roughness is 0.3, and the wind speed and wind direction are related to the input of different conditions below, which have been mentioned below.
(6) Yes, we have deleted some concepts of "common sense" and "science popularization". The length presented so far is basically the findings and conclusions obtained from our research. Make the focus of the article more clear.
Reviewer 2 Report
kindly consider the comment.
· Who is the author? Is a single author assisted by others?
· What is the author’s background? Is the author a PhD student? (see email address)
· Is this paper part of the author's thesis? (see acknowledgement)
This helps me read this paper.
· What is the paper’s contribution to academics, practitioners, policymakers, etc.? Depending on the type of reader, the paper’s topic can be beneficial. However, generally speaking, paper texts as a whole are too vast for readers to digest them. Furthermore, the paper is overflowing with common knowledge, so readers get bored and are tempted to stop reading.
· The paper gives the impression that the authors are more occupied with visual presentation, although visuality has its value.
· References are too few in comparison to long texts.
· Too many subtitles and texts obscure the real outcome of the research in limited paper space. Moreover, if the paper is a rough guide (see the conclusion), what is the point of writing this in quantity and detail?
· The author’s strong voice and critical analysis are missing.
Recommendation
· Verify who the author is. What background?
· Write a short sentence about "acknowledgement". This is not a thesis.
· Delete less-relevant texts and less-important images (30-35%). The paper is too long for the journal. Select only the most pertinent sections and discussions. If desired, the author could explain why other parts of the text had to be deleted, due to the paper or journal or...
· Abstract: Rewrite and shorten the text by adding the aim, objective, methodology, and conclusion. The remaining text from the abstract can be inserted into either the main text or the conclusion.
· Figure 1. If this image is necessary, highlight the area that deals with the topic.
· From line 74: Does this section really deal with the literature review?
· Line 188: "Green Building Design Standards of the Chinese mainland"? Perhaps readers outside of China may not know it. A very short explanation is needed.
· Line 223: Why is the 2009 version of PHOENICS used? Now it is 2023. Each year, technology develops.
· Lines 1211-1222: is this politeness or an excuse or something else? If this is a rough guide, why does the author spend so much time on it, as mentioned above?
· Conclusion: Rewrite.
· Reference: Explain why the reference list is small.
· Grammar: Check the words in between. Delete unnecessary words. Try not to repeat often the same words (lines 344-359: the notion of "significant" cannot be overused all the time).
· Do the image colours come from computer graphics? Colours of strong intensity tend to clash with other images. And readers' eyes get tired of different colour contrasts between various image types.
· Re-formulate texts: Many texts do not need explanation. Keep a balance between texts and images to display academic quality. Quality is more valued than quantity! Less is more.
Author Response
Comments: Who is the author? Is a single author assisted by others?
What is the author’s background? Is the author a PhD student? (see email address).
Is this paper part of the author's thesis? (see acknowledgement)
Response:Thank you for your suggestion.Authors are as indicated in the submitted manuscript. They are: Jialun He, Yile Chen, Liang Zheng, Jianyi Zheng. Among them, Jialun is a master student who has graduated. Yile Chen and Liang Zheng are still PhD students. After Jialun He graduated, the research on wind environment was continued by Yile Chen and Liang Zheng. Jianyi Zheng is the instructor of these three students. In this submission, we all use the school mailbox. The authors are all real. Do not worry. This paper consists of a relatively long study. During the early stages, the first author completed a section and wrote it up as a dissertation. Later, the two PhD students continued their research. Perhaps our acknowledgments were inaccurate. We are very sorry! We have revised the content of this part so far.
Comments: What is the paper’s contribution to academics, practitioners, policymakers, etc.? Depending on the type of reader, the paper’s topic can be beneficial. However, generally speaking, paper texts as a whole are too vast for readers to digest them. Furthermore, the paper is overflowing with common knowledge, so readers get bored and are tempted to stop reading. The paper gives the impression that the authors are more occupied with visual presentation, although visuality has its value. References are too few in comparison to long texts.
Response: We have supplemented it in the section “1.2. Literature Review ”"1.3. Problem Statement and Objectives".We have also added references.
Comments: Too many subtitles and texts obscure the real outcome of the research in limited paper space. Moreover, if the paper is a rough guide (see the conclusion), what is the point of writing this in quantity and detail? The author’s strong voice and critical analysis are missing.
Response: Thank you for your suggestion. Most of the research that has been done so far looks at the wind environment in daily situations. But it's also important to think about how the wind affects buildings when the weather is bad. In this study, the CPD method is used to analyze two different wind environments, daily and extreme, for the high-rise buildings in the ultra-high-density city of Macau. We have made detailed revisions in the text.
Comments: Verify who the author is. What background?
Write a short sentence about "acknowledgement". This is not a thesis.
Delete less-relevant texts and less-important images (30-35%). The paper is too long for the journal. Select only the most pertinent sections and discussions. If desired, the author could explain why other parts of the text had to be deleted, due to the paper or journal or....
Response: Thank you for your suggestion.Authors are as indicated in the submitted manuscript. They are: Jialun He, Yile Chen, Liang Zheng, Jianyi Zheng. Among them, Jialun is a master student who has graduated. Yile Chen and Liang Zheng are still PhD students. After Jialun He graduated, the research on wind environment was continued by Yile Chen and Liang Zheng. Jianyi Zheng is the instructor of these three students. In this submission, we all use the school mailbox. The authors are all real. Do not worry. This paper consists of a relatively long study. During the early stages, the first author completed a section and wrote it up as a dissertation. Later, the two PhD students continued their research. Perhaps our acknowledgments were inaccurate. We are very sorry!We have revised the content of this part so far.
Comments: Abstract: Rewrite and shorten the text by adding the aim, objective, methodology, and conclusion. The remaining text from the abstract can be inserted into either the main text or the conclusion.
Response: Thank you for your suggestion. Our Abstract section has been rewritten. Sentences are condensed than before. Currently 210 words. Structurally references your proposal as well. For details you can see the modified documentation.
Comments: Figure 1. If this image is necessary, highlight the area that deals with the topic.
Response: Thank you for your suggestion.Figure 1 is a display of the historically existing Areia Preta reclamation area, showing the development process from 0 to 1 in this area. In the next article, a comparison of the urban model before and after shows how many people live in high-rise apartment buildings above the city. At present, we have performed secondary processing on the image, and marked the corresponding research area in the image.
Comments: From line 74: Does this section really deal with the literature review?
Response: Thank you for your suggestion. There are directly quoted sentences in this section, which are intended to express the historical reasons and background for the emergence of high-density cities. At the same time, we reorganized the literature review to make the undertaking more logical.
Comments: Line 188: "Green Building Design Standards of the Chinese mainland"? Perhaps readers outside of China may not know it. A very short explanation is needed.
Response: Green building design guidelines on the Chinese mainland: The Chinese Academy of Building Sciences is responsible for the interpretation of specific technical content. Green design applicable to new construction, renovation, and expansion of civil buildings is the most authoritative sustainable design criterion in China. It uses the basic scale, technical requirements, and technical measures as the principles of sustainable design.
Comments: Line 223: Why is the 2009 version of PHOENICS used? Now it is 2023. Each year, technology develops.
Response: Thank you for your suggestion. In fact, the 2009 version of PHOENICS has better compatibility and is also the most stable. At present, many design institutes use this version. However, its version upgrade does not affect the simulation effect of its basic algorithm on the wind environment. The version upgrade mainly relates to the new fluid model, other industry fluid model algorithms, etc., and does not affect the scope of this experiment.
Comments: Lines 1211-1222: is this politeness or an excuse or something else? If this is a rough guide, why does the author spend so much time on it, as mentioned above?
Response: Thank you for your suggestion. At that time, I thought it was some modest statement. Now it seems that it is really not suitable. The dissertation focuses on the research findings. We also experienced a lot in the process of modification. At present, the content of this part has been revised and improved.
Comments: Conclusion: Rewrite.
Response: Thank you for your suggestion. We removed redundant wordy words. It has been streamlined for now.
Comments: Reference: Explain why the reference list is small.
Response: Thank you for your suggestion. We have supplemented it in the section “1.2. Literature Review ”.We have also added references.
Comments: Grammar: Check the words in between. Delete unnecessary words. Try not to repeat often the same words (lines 344-359: the notion of "significant" cannot be overused all the time).
Response: Thank you for your suggestion. We have checked again.
Comments: Do the image colours come from computer graphics? Colours of strong intensity tend to clash with other images. And readers' eyes get tired of different colour contrasts between various image types.
Re-formulate texts: Many texts do not need explanation. Keep a balance between texts and images to display academic quality. Quality is more valued than quantity! Less is more.
Response: Yes, the pictures of simulation results are flat images derived from computer software. Their colors are determined by the software. We can't change it either. We were also considering whether we needed to do Photoshop color adjustment. However, this would be unreal. At the same time, we also refer to papers published in other journals, and they are also images exported directly using computer software. So we also made reservations in this article. In terms of layout, the typesetting may indeed not look good at present. However, more than 30% have been deleted on the basis of the original manuscript. At the same time, some important passages were rewritten.
Round 2
Reviewer 1 Report
All comments are well integrated. Congratulations.
Author Response
Thank you for your affirmation and support! We will try our best to do better in research!
Reviewer 2 Report
Pls, read the comment with an objective attitude.

Author Response
Thank you very much for your patience and advice.
In this revision, we have sorted out the key conclusions of the three experimental simulation processes in the text. As for the specific detailed experimental process and details, as well as the results formed by the simulation software, we have made three appendices.
They are:
Appendix A
Experimental simulation process and results: Wind Environment Simulation at Two Altitudes: Pedestrian Height and Standard Height
Appendix B
Experimental simulation process and results: Influence Analysis of Single Building Form Design Elements in Areia Preta
Appendix C
Experimental simulation process and results: Typhoon Simulation Environment
Our text is 7793 words, 11 illustrations. We have rethought and adjusted things that are too detailed. The text part focuses on analyzing and discussing the relationship between high-rise residential buildings in high-density cities and the wind environment, and obtains the research conclusion of architectural form layout. It also includes the wind environment under severe typhoon weather.
Thank you again.